# Outward open conformation of a Major Facilitator Superfamily multidrug/H$^+$ antiporter provides insights into switching mechanism

Kumar Nagarathinam[1,2,9], Yoshiko Nakada-Nakura [3], Christoph Parthier[2], Tohru Terada [4], Narinobu Juge[5], Frank Jaenecke[1], Kehong Liu[3], Yunhon Hotta[3], Takaaki Miyaji [5], Hiroshi Omote[6], So Iwata[3,7], Norimichi Nomura [3], Milton T. Stubbs[1,2] & Mikio Tanabe[1,8]

Multidrug resistance (MDR) poses a major challenge to medicine. A principle cause of MDR is through active efflux by MDR transporters situated in the bacterial membrane. Here we present the crystal structure of the major facilitator superfamily (MFS) drug/H$^+$ antiporter MdfA from *Escherichia coli* in an outward open conformation. Comparison with the inward facing (drug binding) state shows that, in addition to the expected change in relative orientations of the N- and C-terminal lobes of the antiporter, the conformation of TM5 is kinked and twisted. In vitro reconstitution experiments demonstrate the importance of selected residues for transport and molecular dynamics simulations are used to gain insights into antiporter switching. With the availability of structures of alternative conformational states, we anticipate that MdfA will serve as a model system for understanding drug efflux in MFS MDR antiporters.

[1] ZIK HALOmem, Kurt-Mothes-Straße 3, D-06120 Halle/Saale, Germany. [2] Institut für Biochemie und Biotechnologie, Charles-Tanford-Proteinzentrum, Martin-Luther Universität Halle-Wittenberg, Kurt-Mothes-Straße 3a, D-06120 Halle/Saale, Germany. [3] Department of Cell Biology, Graduate School of Medicine, Kyoto University, Konoe-cho, Yoshida, Sakyo-ku, Kyoto 606-8501, Japan. [4] Agricultural Bioinformatics Research Unit, Graduate School of Agricultural and Life Sciences, The University of Tokyo, 1-1-1 Yayoi, Bunkyo-ku, Tokyo 113-8657, Japan. [5] Advanced Science Research Center, Okayama University, 1-1-1 Kita-ku, Tsushima-naka, Okayama 700-8530, Japan. [6] Department of Membrane Biochemistry, Okayama University Graduate School of Medicine, Dentistry and Pharmaceutical Sciences, 1-1-1 Kita-ku, Tsushima-naka, Okayama 700-8530, Japan. [7] RIKEN, SPring-8 Center, 1-1-1 Kouto, Sayo-cho, Sayo-gun, Hyogo 679-5148, Japan. [8] Structural Biology Research Center, Institute of Materials Structure Science, KEK/High Energy Accelerator Research Organization, 1-1 Oho, Tsukuba, Ibaraki 305-0801, Japan. [9] Present address: Institute of Virology, Hannover Medical School, Carl-Neuberg-Straße 1, D-30625 Hannover, Germany. Correspondence and requests for materials should be addressed to M.T. (email: mikio.tanabe@kek.jp) or to M.T.S. (email: stubbs@biochemtech.uni-halle.de)

fflux transport of antibiotics and other potentially harmful compounds from the bacterial cytoplasm by multidrug resistance (MDR) transporters represents an increasing challenge for the treatment of pathogenic bacterial infection[1–3]. A large number of MDR transporters belong to the Major Facilitator Superfamily (MFS), found in both Gram-positive and -negative organisms[1,2]. Typical MFS transporters possess 12 transmembrane helices (TMs) divided into two pseudo-symmetrical 6TM N- and C-terminal lobes. Changes in relative orientation of the two lobes within the plane of the bilayer (the rocker-switch mechanism[4]) allow alternating access to the cytoplasmic and extracellular/periplasmic sides of the membrane, facilitating directed transport of substrates across the membrane, with the transporter cycling between outward open ($O_o$), inward open ($I_o$) and intermediary occluded states[5–7]. Despite progress in structural determinations of these states for uniporter and symporter MFS transporters, few such data are available for antiporters.

MdfA, an MFS-MDR transporter from *E. coli* with homologs in many pathogenic bacteria, is an extensively characterized drug/$H^+$ antiporter[8]. It transports lipophilic, cationic, and neutral substrates, in each case driven by the proton motive force[9,10]. Two acidic residues within TM1, $Glu26^{TM1}$ and $Asp34^{TM1}$, have been implicated in proton ($H^+$) and substrate transport coupling[11–13], and it has been proposed that changes in their protonation could lead to local structural changes within the binding pocket upon $H^+$/substrate binding[11]. The recently reported structure of chloramphenicol-bound MdfA in an inward facing ($I_f$) conformation[14] reveals the antibiotic bound in the immediate vicinity of $Asp34^{TM1}$, in line with earlier biochemical data[12,13].

In order to gain a complete picture of the efflux mechanism, however, structural data for alternative states are required. Here we report the crystal structure of MdfA in the $O_o$ state and identify conformational changes that accompany transitions between the $I_f$ and $O_o$ states. With the availability of structures of alternative conformational states, we anticipate that MdfA will serve as a model system for understanding drug efflux in MFS MDR antiporters.

## Results

**Overall structure of MdfA in the outward open ($O_o$) state.** The crystal structure of Fab-bound MdfA presented here reveals the transporter in the outward open ($O_o$) state, with the N- and C-lobes approaching each other closely at the intracellular face of the transporter (Fig. 1). The N-terminus of TM5 juxtaposes the C-termini of TM8 and TM10 and the N-terminus of TM11 nestles between the C-termini of TM2 and TM4. Access to the transporter cavity from the cytoplasmic face is sealed off by formation of a hydrophobic plug through intercalation of side-chains from each of these helices centered around $Phe340^{TM10}$ (Fig. 2). These contacts are supported by mutually favorable interactions between the side chain of $Arg336^{TM10}$ and the C-terminal dipole of TM4, and $Asp77^{TM2}$ and the N-terminal dipole of TM11. $Asp77^{TM2}$ (from conserved motif A) is in addition part of an electrostatic cluster involving $Arg81^{TM3}$ and $Glu132^{TM5}$, with an adjacent cluster including $Arg78^{TM2}$ and residues of the intermediate loop ($Arg198^{\alpha6-7}$) and helix ($Asp211^{\alpha6-7}$).

In the ligand bound $I_f$ state, in which the two lobes rotate largely as rigid bodies by 33.5° about an axis parallel to the plane of the membrane bilayer, these multiple interactions are replaced by predominantly hydrophobic contacts between the periplasmic halves of TMs 1, 2, and 5 of the N-terminal domain and TMs 7, 8, and 11 of the C-terminal domain. This is effected by a sliding of TM11 along TM2 and a significant rearrangement of TM5 (see below), closing the transporter cavity to the periplasm. The drug binding pocket observed in the $I_f$ state is disrupted in the $O_o$ state through displacement of $Ala150^{TM5}$ and $Leu151^{TM5}$ (see below) as well as lateral movement of C-terminal domain residues from TMs 7 and 8 (Supplementary Fig. 1).

**MdfA helix TM5 is kinked and twisted in the $O_o$ state.** Superposition of the individual domains of MdfA in the $I_f$ and $O_o$ conformations reveals significant deviations in the N-terminal domain (Supplementary Fig. 2). The largest of these are

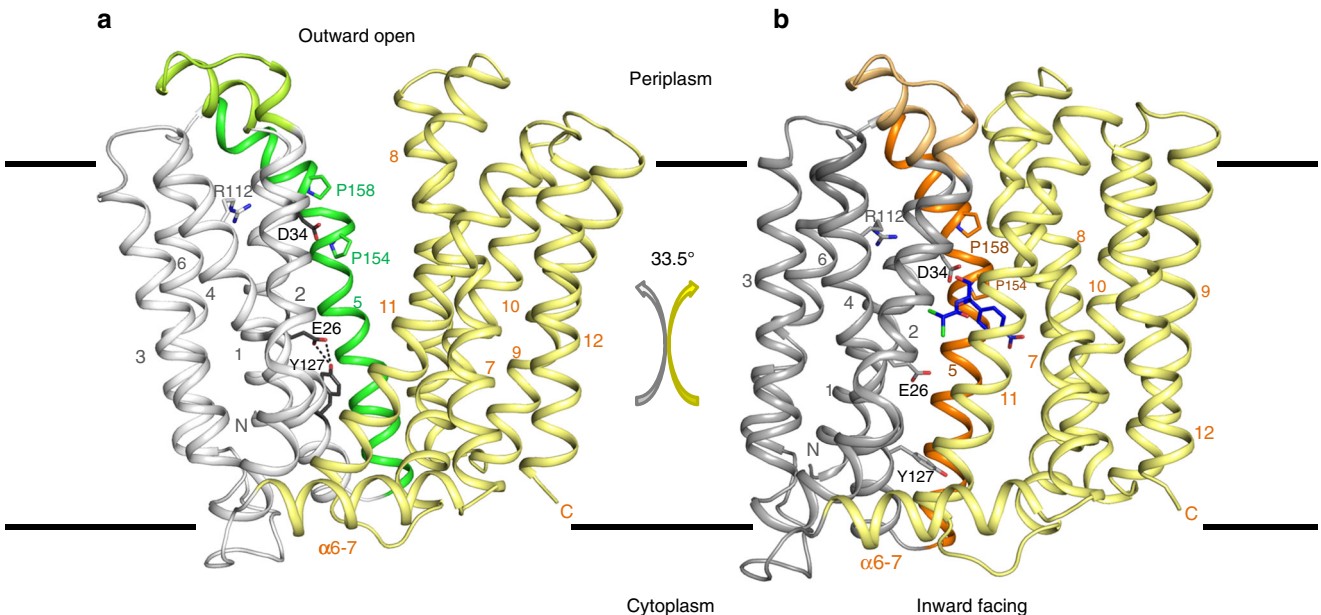

**Fig. 1** Overall structure of MdfA in the outward open ($O_o$) and inward facing ($I_f$) states. **a** The transporter in the $O_o$ conformation (this work); **b** MdfA in the ligand-bound $I_f$ state (ref. [14]). The N- (white/gray) and C-terminal (yellow) six transmembrane helical domains are shown in ribbon representation, with transmembrane helices (TMs) numbered. Note the difference in relative orientation of the two domains by 33.5°. TM5, whose conformation differs between the two states, is shown in green ($O_o$) or orange ($I_f$); the TM1–TM2 termini are in corresponding light colors. The position of chloramphenicol bound in the $I_f$ state is depicted using blue sticks

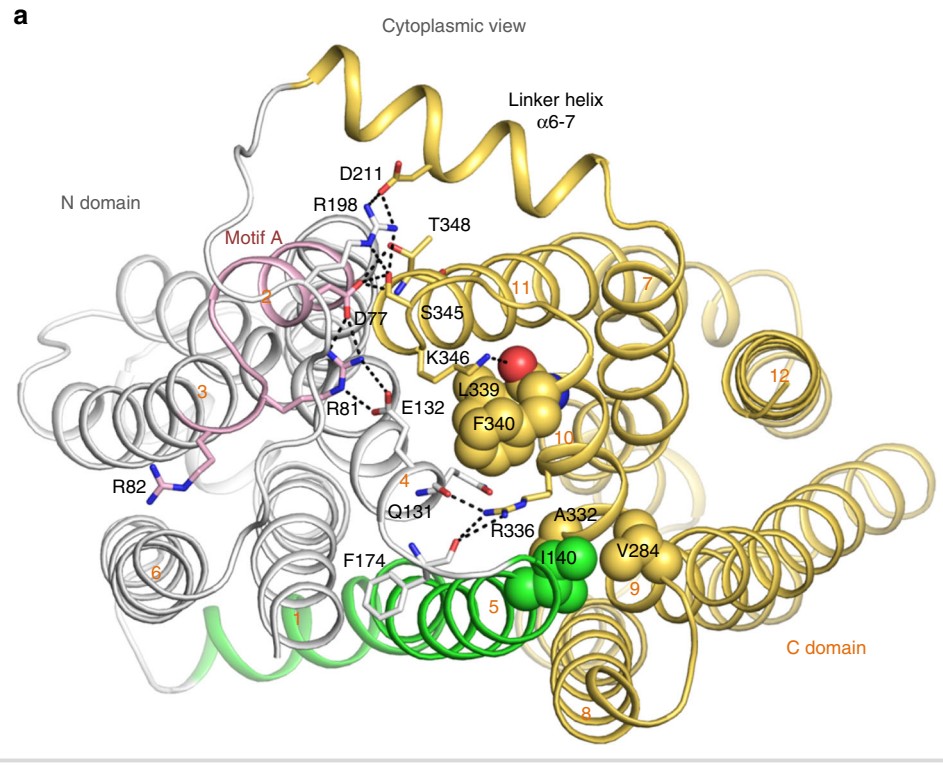

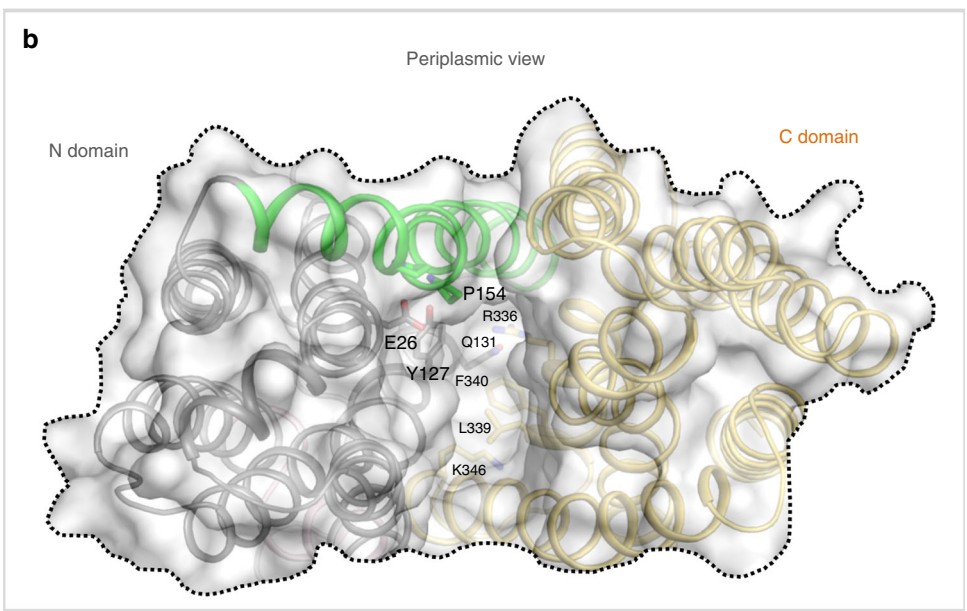

**Fig. 2** Cytoplasmic and periplasmic faces of MdfA in the outward open conformation. **a** The cytoplasmic entrance to the ligand-binding pocket is closed in the $O_o$ conformation by numerous interactions between the N- and C-lobes (view obtained by rotating Fig. 1 90° about a horizontal x-axis). The N-terminus of TM5 juxtaposes the C-termini of TM8 and TM10, and the N-terminus of TM11 nestles between the C-termini of TM2 and TM4. Hydrophobic sidechains from each of these helices pack against each other to form a hydrophobic plug that seals off access to the transporter cavity from the cytoplasmic face, supported by additional mutually favorable electrostatic interactions. **b** View from the periplasmic face (following a 180° rotation about a horizontal x-axis), demonstrating the deep cavity between the two domains in the outward open conformation. Dotted line denotes approximate boundary delineated by the bacterial membrane outer leaflet head groups

found in TM5, which ends in the antiporter motif C 153AlaProXaaXaaGlyPro158 that is absent in symporters and uniporters[15]. Whereas TM5 in the $I_f$ structure adopts an α-helical conformation of almost ideal geometry up to motif C, residues 136 to 153 in the $O_o$ structure exhibit a profound 15° kink, accompanied by a ca. 45° clockwise twist parallel to the helix axis that terminates with the two-proline-containing motif C (Fig. 3; Supplementary Fig. 3; Supplementary Movie). This results in a

repositioning of the hydrophobic side chains Ile142[TM5], Leu145[TM5], Met146[TM5], and Val149[TM5] with respect to the N-terminal domain core. Leu145[TM5], which in the $I_f$ conformation associates with the N-terminal domain, engages instead with residues of the C-terminal domain in the $O_o$ state. The carbox-amide of conserved Asn148[TM5] is removed from a (presumably hydrophobic) membrane exposed location in the $I_f$ state to form hydrogen bonds with the side chain of Asn272[TM8] and the main

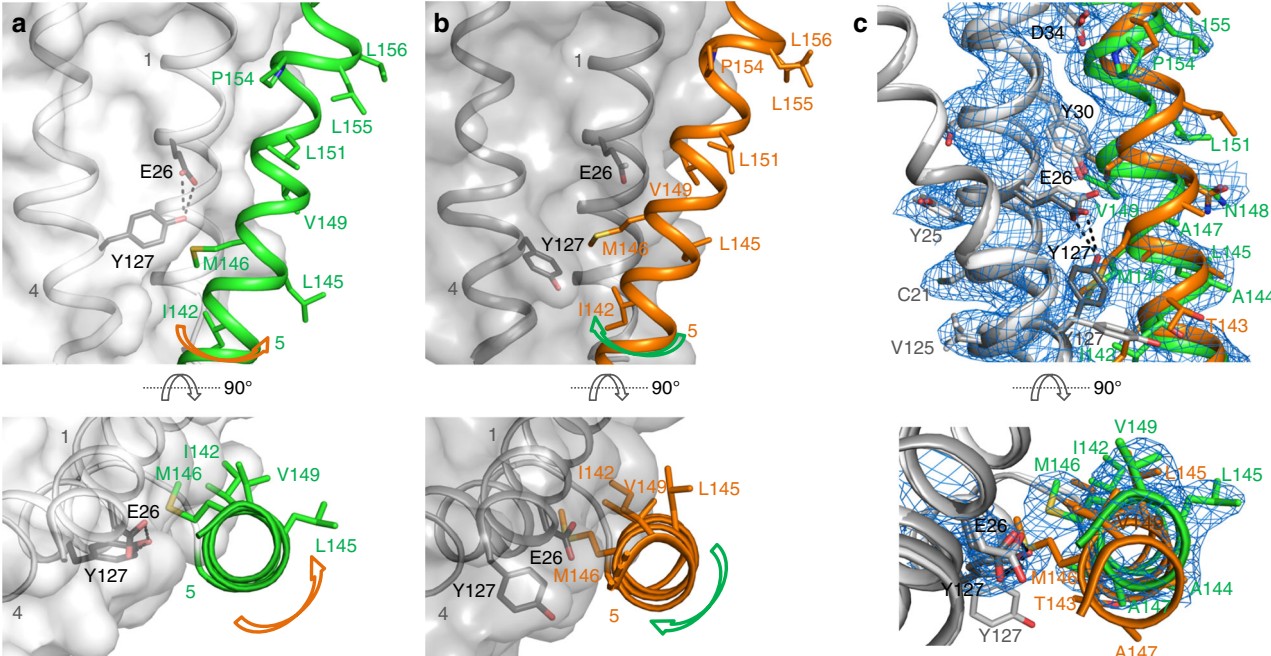

**Fig. 3** The $O_o$ and $I_f$ conformations differ by local twisting of TM5. **a** In the $O_o$ state, TM5 (green) in the N-terminal domain is partially distorted, resulting in Cα displacements compared to the $I_f$ state of up to 2.9 Å (Met146$^{TM5}$). The side chain of Met146$^{TM5}$ rests against the phenolic side chain of Tyr127$^{TM4}$, whose hydroxyl moiety is ca. 2.5 Å from the side chain carboxylate group of Glu26$^{TM1}$. **b** TM5 adopts an almost ideal α-helical conformation in the $I_f$ state through displacement of the Tyr127$^{TM4}$ side chain by that of Met146$^{TM5}$. TM5 straightens, rotating around its axis such that its hydrophobic side chains can engage/disengage the C-terminal domain. **c** Electron density for TM5 in the $O_o$ conformation, superimposed with coordinates of the final (green) and initial (orange) models. See also Supplementary Fig. 3 and Supplementary Movie

chain carbonyl of Ile269$^{TM8}$ in the $O_o$ conformation. The largest deviation in TM5 is found at residue Met146$^{TM5}$, the side chain of which rests against the aromatic moiety of Tyr127$^{TM4}$ in the present structure. In turn, the side chain hydroxyl of Tyr127$^{TM4}$ is found within hydrogen bonding distance (2.5 Å) of the carboxylate of Glu26$^{TM1}$ in the $O_o$ structure. Crucially, the space occupied by the Tyr127$^{TM4}$ side chain in the $O_o$ state is in the $I_f$ structure replaced by that of Met146$^{TM5}$.

**Rearrangements in the N-terminal domain hydrophobic core.** Also of note are small yet significant changes in the region Leu41$^{TM1}$–Val54$^{TM2}$, which runs from the C-terminus of TM1 and the N-terminus of TM2 (Supplementary Fig. 4). The side chains of residues Val43$^{TM1}$, Tyr47$^{TM1}$, and Trp53$^{TM2}$ form part of a hydrophobic cluster near the periplasmic face of MdfA that includes Met40$^{TM1}$, Ile105$^{TM4}$, and Phe108$^{TM4}$ from TM4, and Trp170$^{TM6}$ and Phe174$^{TM6}$ at the N-terminus of TM6. This cluster, which juxtaposes the buried guanidinium moiety of Arg112$^{TM4}$ that is absolutely conserved among MFS homologs (motif B), exhibits a small but significant expansion in the present structure compared to that in the $I_f$ conformation. While the structural differences may appear to be small, we note that the transporter MdfA is stabilized by ligand binding[14,16], so that the transporter in the unbound $I_o$ state could well differ from the ligand bound MdfA $I_f$ structures presented by Heng et al.[14,17].

In order to test the effect of amino acid substitutions on chloramphenicol transport experimentally, MdfA and its variants were reconstituted in proteoliposomes following procedures described for the chloroquine resistance transporter from *Plasmodium falciparum*[18] (Fig. 4). Purified reconstituted wild-type MdfA was able to transport 50 pmol chloramphenicol (per mg protein per minute), which compares favorably with the 3 pmol mg$^{-1}$ min$^{-1}$ determined using crude membrane preparations[19], whereas transport proved unaffected by mutation of

Glu26$^{TM1}$ to Gln, suggesting that the charge state of this residue is not crucial for chloramphenicol transport. The variants Tyr127$^{TM4}$Phe, Met146$^{TM5}$Ala, and Trp170$^{TM6}$Ala of the hydrophobic core all showed significant reductions in transport in the presence of a pH gradient. As expected, chloramphenicol transport was low in the absence of ΔpH, arising from downhill transport due to the initial infinite substrate gradient.

**Molecular dynamics simulations.** To gain further insights into the transport cycle, molecular dynamics (MD) simulations were performed with all possible combinations of Glu26$^{TM1}$ and Asp34$^{TM1}$ protonation states, starting from either the $O_o$ structure without Fab or the $I_f$ structure without chloramphenicol. Initial trials with the present structure assuming both acidic residues to be deprotonated [$O_o$(E26$^-$/D34$^-$)] indicated that the overall structure remained unchanged during the MD simulation. The C-terminal lobe is more rigid than the N-terminal lobe, and the cytoplasmic halves of the TM helices (relative to the center of the lipid bilayer) showed smaller root mean square deviation (RMSD) values than those of the periplasmic halves of the TM helices (Supplementary Table 1)—despite the fact that the Fab binds to the cytoplasmic face of MdfA. We therefore conclude that the outward-open conformation of MdfA is little affected by Fab binding (corroborated by low-resolution data from crystals of MdfA alone[20]) and is stably maintained in a solvated membrane environment.

To monitor the degree of conformational change, two distances were used to describe the opening and closing of the periplasmic and cytoplasmic sides of the transporter ($d_1$ and $d_2$ respectively) (Fig. 5). Starting from the $O_o$ state, the largest divergence from the crystal structure was observed when Asp34$^{TM1}$ was protonated [$O_o$(E26$^-$/D34$^P$) and $O_o$(E26$^P$/D34$^P$)], resulting in small values of $d_1$ and $d_2$. This state corresponds to an occluded conformation, as both the periplasmic and cytoplasmic entrances

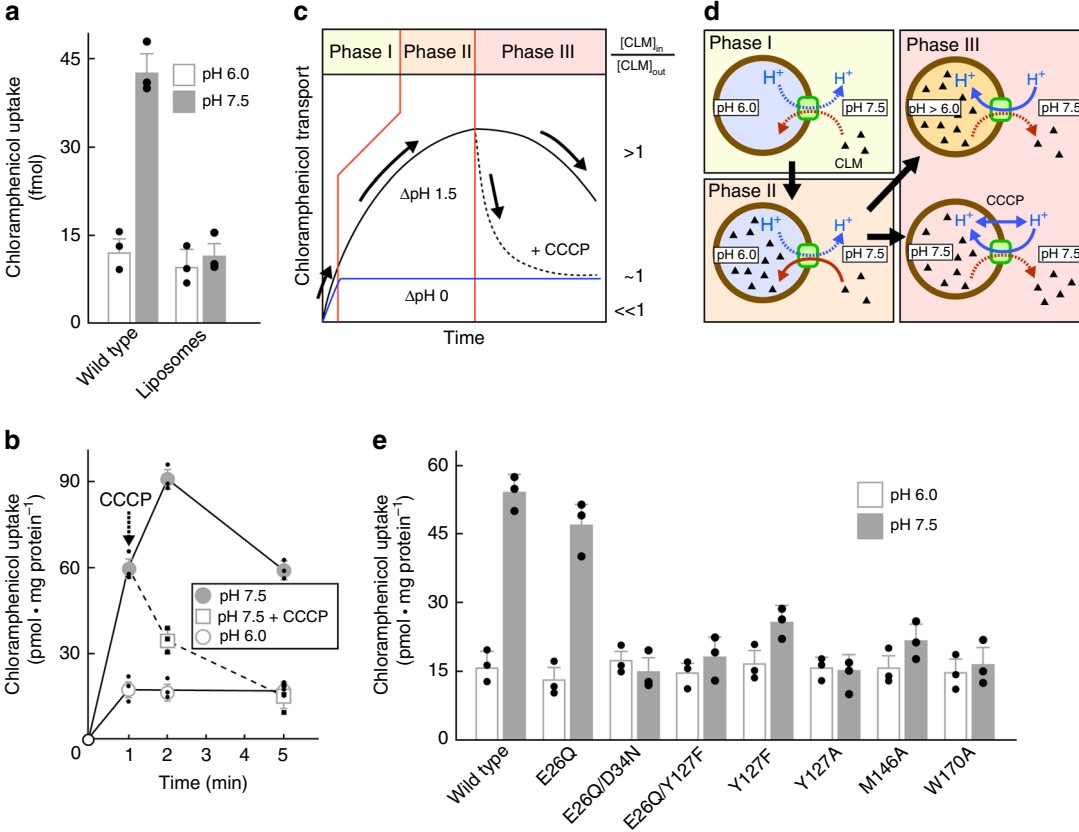

**Fig. 4** Chloramphenicol transport by MdfA reconstituted in proteoliposomes. **a** Chloramphenicol transport into reconstituted proteoliposomes is dependent upon the presence of MdfA and a pH gradient. **b** Time course for uptake using reconstituted MdfA. In the absence of a pH gradient (open circles), downhill-like transport (with the substrate gradient) occurs rapidly due to the small volume of the proteoliposomes. In the presence of a pH gradient, however, chloramphenicol uptake (filled circles) involves at least three phases: following a rapid initial downhill transport phase (not visible), uphill accumulation of the substrate in the liposomal lumen against the concentration gradient takes place at the expense of proton export (II). Within a few minutes, the situation is reversed due to lumen acidification, leading to chloramphenicol efflux (phase III). Crucially, collapse of the pH gradient through administration of the $H^+$-ionophore CCCP (open squares) results in rapid chloramphenicol efflux (downhill transport) until the luminal concentration reaches that observed in the absence of a pH gradient. **c, d** Schematic diagram illustrating the phases of chloramphenicol (CLM) uptake in the reconstituted system. **e** Uptake by proteoliposomes containing purified MdfA variants in the presence (closed bars) and absence (open bars) of a pH gradient at 1 min. Data are mean values ± s.d., $n = 3$

are closed. Analysis of the free energy landscape for this transition (Supplementary Fig. 5a) indicates that upon protonation of Asp34$^{TM1}$, the $O_o$ state is much less stable than the occluded state, suggesting that the transition occurs rapidly and is in effect irreversible. During the simulations, the hydrogen bond between Glu26$^{TM1}$ and Tyr127$^{TM4}$ was for the most part maintained, although the carboxylate at times also hydrogen bonded to Tyr30$^{TM1}$ (Supplementary Fig. 6).

In the occluded state adopted following Asp34$^{TM5}$ protonation, the acidic side chain juxtaposes an internal cavity bounded by the conserved residues Tyr257$^{TM8}$, Gln261$^{TM8}$ as well as the hydrophobic side chains of Ile239$^{TM7}$ and Phe265$^{TM8}$; this cavity is also observed in the $I_f$ crystal structure (Supplementary Fig. 7). TM5 is straighter than in the $O_o$ and $I_f$ structures, although it adopts a twisted conformation as in the $O_o$ structure.

Starting from the $I_f$ crystal structure after removal of the ligand and replacement of Arg131$^{TM4}$ by the wild-type Gln, a similar occluded state was obtained during the MD run $I_f$(E26$^-$/D34$^P$). In contrast to the transition from the $O_o$ state, however, the $I_f$ and the occluded states are in a flat free-energy well (Supplementary Fig. 5b), suggesting that these two states can co-exist when Asp34$^{TM5}$ is protonated and that the transition between the $I_f$ state and the occluded state is reversible. As we observed only a one-way transition from the $I_f$ state to the occluded state in the

1 µs MD simulation, the transition must be slow, presumably due to the complex and rugged nature of the original multi-dimensional energy surface. During this simulation, TM5 twisting was observed due to the close approach of the cytoplasmic halves of TM5 and TM8 (i.e. decreasing $d_2$), following accommodation of Leu151$^{TM5}$ in the space between Leu268$^{TM8}$ and Ile269$^{TM8}$ (Supplementary Fig. 7). During the simulations where Glu26$^{TM1}$ was protonated [$I_f$(E26$^P$/D34$^-$) and $I_f$(E26$^P$/D34$^P$)], Tyr127$^{TM4}$ moved toward Glu26$^{TM1}$ (Supplementary Fig. 6).

## Discussion

The structure of MdfA presented here reveals features of the $O_o$ conformation and allows comparison with the previously determined ligand bound $I_f$ state[14]. Going from the $I_f$ to the $O_o$ state, the N- and C-terminal domains of the transporter reorient in the membrane largely as rigid bodies, with the exception of three regions: (i) transmembrane helix TM5 kinks and twists, (ii) the periplasm-proximal hydrophobic core of the N-terminal domain reorganizes, and (iii) a cytoplasmic loop of the C-terminal domain rearranges to accommodate closure of the cytoplasmic entrance (see Supplementary Movie). The twisting of the helix in the $O_o$ conformation appears to be prevented from transiting to a straight-form as seen in the $I_f$ state by juxtaposition of the

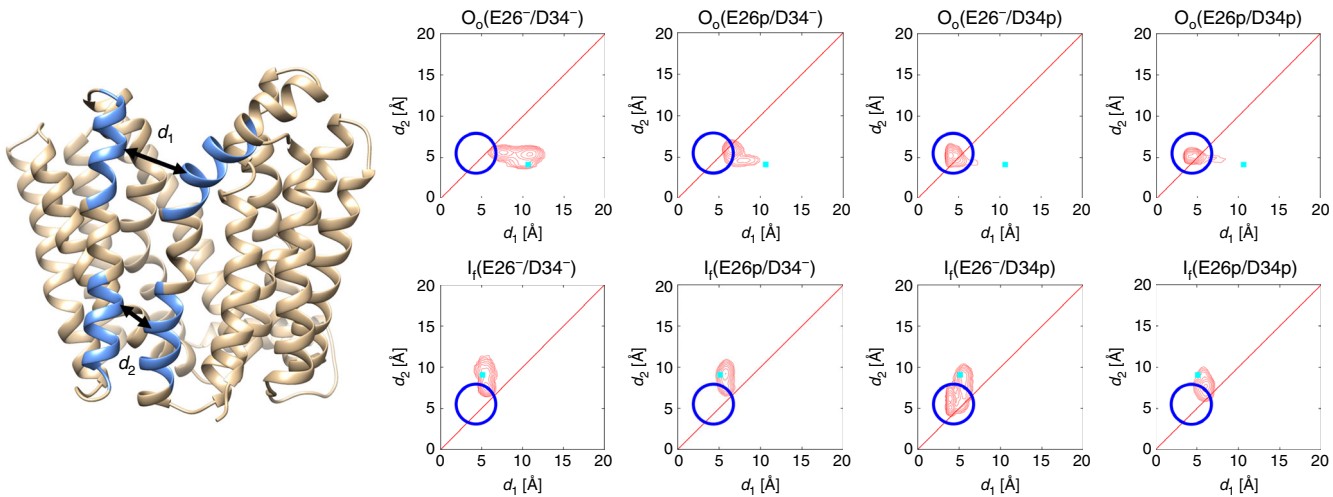

**Fig. 5** Conformational distributions of MdfA obtained following MD simulations. Starting from each initial conformation ($O_o$ vs. $I_f$) and Glu26/Asp34 protonation state, the conformational distributions of the MD simulations were calculated as a function of $d_1$ and $d_2$ ($d_1$: minimum distance between $C\alpha$ atoms of residues 156–165 (TM5) and those of residues 253–262 (TM8); $d_2$: minimum distance between $C\alpha$ atoms of residues 139–148 (TM5) and those of residues 270–279 (TM8)). Cyan squares indicate the corresponding distances in the initial conformations ($O_o$: this study; $I_f$ PDB 4ZOW), and the blue circles indicate the position of the peak in the plot for $O_o$(E26$^-$/D34$^p$)

Tyr127$^{TM4}$ and Met146$^{TM5}$ side chains, with the aromatic side chain hydroxyl of Tyr127$^{TM4}$ held in place by a hydrogen bond to Glu26$^{TM1}$. Our reconstitution experiments demonstrate the importance of Tyr127$^{TM4}$ and Met146$^{TM5}$ for transport, and suggest that the charge state of Glu26$^{TM1}$ is of little significance for chloramphenicol transport in the presence of a pH gradient, which is consistent with previous results[19]. It should be noted that, strictly speaking, the conclusions presented here apply only to chloramphenicol transport (for which structural data of the $I_f$ form are available); whereas we expect them to be generally valid for other neutral MdfA substrates, the situation may differ for other substrates.

To gain further insights into the transport process, we performed MD simulations involving different protonation states of the two acidic residues identified previously as being important in in vivo studies[11,16,21], Asp34$^{TM1}$ and Glu26$^{TM1}$. Within the timescale of our simulations, protonation of Asp34$^{TM1}$ through exposure to the low pH periplasmic space leads to an occluded state in which the acidic side chain becomes enclosed in an internal cavity that recapitulates its environment in the $I_f$ conformation. TM5 continues to be twisted in this occluded state and the Glu26$^{TM1}$–Tyr127$^{TM4}$ hydrogen bond remains intact, although the charge state of Glu26$^{TM1}$ does not appear to play a role in this. Nevertheless, previous in vivo studies have shown that Glu26$^{TM1}$ is critical for the transport of cationic substrates[11,16], so that the situation may be different for cationic and lipophilic substrates, in which the initial $I_f$ state assumed here might not apply.

MD simulations also demonstrate that an occluded state and a twisted TM5 conformation can be obtained starting from the $I_f$ state. The fact that the transporter is stabilized by ligand binding[14,16] however means that the $I_f$ structures presented by Heng et al.[14,17] might not provide an accurate representation of MdfA in the unbound $I_o$ state (moreover, these ligand-bound structures were obtained using a mutated variant Gln131$^{TM4}$Arg, which has recently been reported to be transport inactive[22,23]). Thus TM5 untwisting might occur on going from the occluded to the $I_o$ state, or upon ligand binding to form the $I_f$ state.

Recent structure determinations of other transporters[5–7] indicate that individual helices within each domain can exhibit significant variability upon conformation switching. For MdfA, the existence of structurally underdetermined $I_o$ and intermediate occluded states (whereby ligand bound and unbound occluded states are likely to be different) thus precludes a detailed description of the complete transport cycle. Nevertheless, the combination of data presented here suggests an important role for the interaction between Glu26$^{TM1}$ and Tyr127$^{TM4}$. We note that through their common location on helix TM4, the orientation of the Tyr127$^{TM4}$ side chain could couple with the environment of the motif B Arg112$^{TM4}$ side chain. The buried guanidinium moiety is involved in an elaborate hydrogen bonding network involving the even more buried Gln115$^{TM4}$, the carbonyl carbon of Gly32$^{TM1}$, and (via a solvent molecule identified in the high-resolution structure[14]) Asn33$^{TM1}$ and Asp34$^{TM1}$. Residues Arg112$^{TM4}$, Asp34$^{TM1}$, Gln115$^{TM4}$, and Gly32$^{TM1}$ have all been shown to be important for MdfA action[11,14,16,19,24], and a role of the surrounding hydrophobic cluster is confirmed by the deleterious effect on chloramphenicol transport of the Trp170$^{TM6}$Ala mutation. Changes in the chemical environment of Asp34$^{TM1}$ (e.g. by ligand binding or changes in protonation) could therefore lead to the observed reorganization of the hydrophobic cluster immediately adjacent to Arg112$^{TM4}$. In turn, communication of this change through TM4 could influence the orientation of the Tyr127$^{TM4}$ side chain, dictating the position of that of Met146$^{TM5}$ and thereby the degree of TM5 twist. Releasing the twist of TM5 from the $O_o$ to the $I_f$ conformation would result in a repositioning of the hydrophobic side chains Ile142$^{TM5}$, Leu145$^{TM5}$, Met146$^{TM5}$, and Val149$^{TM5}$ with respect to the N-terminal domain core, allowing Leu145$^{TM5}$ to dissociate from the N-terminal domain to engage the C-terminal domain.

Support is provided by MdfA rescue mutants. Selection for drug transport rescue in cells harboring the otherwise inactive TM1 variants Glu26$^{TM1}$Thr/Asp34$^{TM1}$Met and Glu26$^{TM1}$Thr resulted in the detection of mutants containing the acidic side chains Ala150$^{TM5}$Glu and Val335$^{TM10}$Glu[11,25]. These residues would be well positioned to make hydrogen bonds to Tyr127$^{TM4}$ in the outward open structure (Supplementary Fig. 8). Recent thermodynamic calculations and molecular dynamic simulations have led in principle to similar conclusions for the L-fucose/H$^+$ symporter FucP[26]. Using computational methods, it was

**Table 1 Data collection and refinement statistics**

| | MdfA–Fab YN1074 |
|---|---|
| **Data collection** | |
| Space group | $P6_122$ |
| Cell dimensions | |
| $a, b, c$ (Å) | 73.26, 73.26, 927.92 |
| $\alpha, \beta, \gamma$ (°) | 90.00, 90.00, 120.00 |
| Resolution (Å) | 49–3.4 (3.61–3.4)[a] |
| $R_{sym}$ or $R_{merge}$ | 25.4 (166.9) |
| $I / \sigma I$ | 11.57 (1.59) |
| Completeness (%) | 99.9 (99.9) |
| Redundancy | 17.3 (16.03) |
| **Refinement** | |
| Resolution (Å) | 49–3.4 |
| No. of reflections | 22216 |
| $R_{work}/R_{free}$ | 25.8/28.3 |
| No. of atoms | |
| Protein | 6,134 |
| Ion | 5 |
| $B$-factors (Å$^2$) | |
| Protein | 113.3 |
| Ion | 136.8 |
| R.m.s. deviations | |
| Bond lengths (Å) | 0.003 |
| Bond angles (°) | 0.596 |

[a]Values in parentheses are for highest-resolution shell

proposed that protonation of FucP Glu135[TM4] in TM4 allows surmounting of a ca. 2 kcal mol$^{-1}$ energy barrier between the inward and outward open states. An intermediate state in which TM11 is distorted is postulated, although a causative link between Glu135[TM4] (de)protonation and TM11 distortion has not been described. Inspection of the FucP structure[27] suggests that Glu135[TM4] could form a hydrogen bond with Tyr365[TM10] of C-terminal domain TM10 (Supplementary Fig. 8). Interestingly, C-terminal domain TM11 is the counterpart of TM5 in the (inverted topology) N-terminal domain[28], reflecting the pseudo-symmetry of the two domains, so that the antiporter MdfA and symporter FucP might be thought of as examples of repeat swapping to yield similar transport mechanisms.

The presence of the MFS-antiporter motif C would appear to be central to transporter switching—restricting the twist of TM5 to a small localized helical segment to facilitate relative rotation of the two domains, and transmitting these perturbations to an adjacent pliable hydrophobic cluster. As other structurally well-characterized antiporter families (such as the amino acid/poly-amine/organocation (APC) transporter superfamily[29] and cation/H$^+$ antiporter family[30]) have been shown to utilize other mechanisms, this may be a property specific to MFS-antiporters. The $O_o$ structure presented here could serve as a template for the design of novel MFS inhibitors that are able to access their target directly from the bacterial exterior.

## Methods

**Crystal structure solution.** Isolation of Fab fragments that recognize native conformations of MdfA in proteoliposomes has been described elsewhere[31]. Co-crystals of MdfA in complex with Fab fragments grew in the lipidic cubic phase and diffracted to 3.4 Å resolution[20,31]. Diffraction data were collected at 100 K at a wavelength of 1.0 Å on the SLS beamline PXI (X06SA) using the 16M Eiger detector. In order to resolve the very long $c$-axis (929 Å), the crystal was mounted so that the rotation axis was ca. 30° to the crystallographic $c$-axis. The synchrotron beam (with beam size increased from 10 to 100 µm) was defocused from the crystal to the detector. Data sets (180° in 0.1° steps) were processed using the program XDS[32]. The crystal belongs to the space group $P6_122$ with one complex in the asymmetric unit. Phases were determined by molecular replacement with PHASER MR[33] using the separated N- and C-lobes of MdfA in the inward open conformation (PDBID: 4ZP0)[14] and an Fab fragment (PDBID:1IBG)[34] as individual

search models. The replacement model was rebuilt manually using COOT[35] and refined using PHENIX[36] with TLS refinement (three groups per polypeptide chain) to an $R_{free}$ value of 28.3%. The final model consists of MdfA residues Gln14–Lys400, Fab heavy chain residues Leu4–Pro216, and Fab light chain residues Asp1–Arg211, as well as one sulfate ion. Ramachandran analysis demonstrates that 94.4% of the residues are in the favorable regions and 5.5% in the allowed regions. One residue (Ser65 in the Fab fragment heavy chain) was in the outlier regions. Atoms for difference density corresponding to solvent molecules, lipids, and/or detergents were not added to the model in accordance with the low resolution of the data. The Fab binds to the cytoplasmic side of MdfA, where it may stabilize the outward open state and enhances crystal contacts[20]. Data collection and refinement statistics are given in Table 1. RMSDs in Cα in positions between $O_o$ and $I_f$ states (Supplementary Fig. 2) were calculated as a function of residue number after separate superposition of the N- and C-terminal domains using the programme LSQKAB from the CCP4 suite[37]. The kink of α-helix TM5 was calculated using Kink Finder[38]. Figures and movies were prepared using PyMOL (Schrödinger, LLC).

**Reconstitution of MdfA.** MdfA mutants were generated using a PCR site-directed mutagenesis kit (Agilent) with the primers listed in the Supplementary Table 2, and purified as for wild-type MdfA. Forty micrograms of purified wild type or mutant MdfA was mixed with 500 µg of azolectin liposomes (Sigma type IIS), frozen at 193 K for at least 10 min[18]. The mixture was thawed quickly by holding the sample tube in the hand and diluted 60-fold with reconstitution buffer containing 20 mM MES-NaOH (pH 6.0), 0.1 M sodium chloride, 5 mM magnesium chloride. Reconstituted proteoliposomes were pelleted by centrifugation at 200,000 × g at 277 K for 1 h, and suspended in 0.2 mL of same buffer.

**Transport assay.** The transport assay mixture (0.2 mL) containing 20 mM MOPS-Tris (pH 7.5) or 20 mM MES-NaOH (pH 6.0), 0.1 M sodium chloride, 5 mM magnesium chloride, and 2 µM [ring-3,5 $^3$H] chloramphenicol (0.5 MBq µmol$^{-1}$, PerkinElmer) was incubated at 300 K for 3 min. Proteoliposomes containing MdfA (0.5 µg protein per assay) (or liposomes as control) were added to the mixture to initiate transport, and incubated for a further 1 min. Aliquots (130 µL) were taken and centrifuged through a Sephadex G-50 (fine) spin column at 760 × g for 2 min. Radioactivity in the eluate was counted using a liquid scintillation counter (PerkinElmer). As a control, the ionophore carbonyl cyanide m-chlorophenyl hydrazone (CCCP) was added to the assay (final concentration 1 µM) after 1 min to collapse the pH gradient. Bovine serum albumin was used as a protein concentration standard[39].

**MD simulations.** Initial coordinates of MdfA in the $O_o$ conformation were taken from those of the crystal structure of the MdfA–Fab complex. N- and C-terminal MdfA residues were capped with acetyl and N-methyl groups, respectively. All histidine residues were protonated on the Nδ1 atom. All acidic residues (excluding Glu26[TM5] and Asp34[TM5]) and all basic residues were deprotonated and protonated, respectively (see Supplementary Table 3). Glu26[TM5] was protonated in the initial structure of the $O_o$(E26$^P$/D34$^-$) and $O_o$(E26$^P$/D34$^P$) simulation runs and Asp34[TM5] was protonated for the $O_o$(E26$^-$/D34$^P$) and $O_o$(E26$^P$/D34$^P$) simulation runs. After filling the large cavity on the periplasmic side of the protein with water molecules to prevent placement of lipid molecules there, the structure was embedded in a lipid bilayer and solvated with water and ions using CHARMM-GUI[40]. The orientation of MdfA relative to the lipid bilayer was determined analogously to that of MdfA in the $I_f$ state (PDB ID: 4ZP0) as deposited in the Orientations of Proteins in Membranes (OPM) database[41] by alignment of the Cα atoms of residues 203–400. The rectangular simulation system generated by CHARMM-GUI (90.76 Å × 90.76 Å × 96.98 Å) was subjected to periodic boundary conditions. The system was composed of one protein, 223 1-palmitoyl-2-oleoyl-sn-phosphatidylethanolamine (POPE), 40 or 41 K$^+$, 44 or 45 Cl$^-$, and about 14,000 water molecules. The CHARMM36 force-field parameters[42,43] were used for protein, lipid, and ions, and the TIP3P model[44] was used for water. Initial coordinates for the simulations starting from the $I_f$ conformation were generated in a similar manner. Here, the coordinates of the chloramphenicol-bound, $I_f$ form of MdfA (PDB ID: 4ZOW) was used after the ligand coordinates were removed and Arg131 was replaced with the wild-type Gln residue. The system size was 92.01 Å × 92.01 Å × 98.11 Å and was composed of 233 POPE, 41 or 42 K$^+$, 47 or 48 Cl$^-$, and about 14,700 water molecules.

After energy minimization and equilibration, a production MD run was performed for 1.6 µs (starting from the $O_o$ conformation) or 1.0 µs (starting from the $I_f$ conformation). During the simulation, the temperature was kept at 303.15 K using the Nosé–Hoover thermostat[45,46], and the pressure was kept at $1.0 \times 10^5$ Pa using the semi-isotropic Parrilello-Rahman barostat[47,48]. Bond lengths involving hydrogen atoms were constrained using the LINCS algorithm[49,50] to allow the use of a large time step (2 fs). Electrostatic interactions were calculated with the particle mesh Ewald method[51,52]. All MD simulations were performed with Gromacs 5.0.5 (ref. [53]), with coordinates recorded every 10 ps.

Time evolutions of RMSDs in the $O_o$(E26$^-$/D34$^P$) and $I_f$(E26$^-$/D34$^P$) MD runs were calculated using their respective initial and final structures as reference (Supplementary Fig. 9). The average values of the RMSDs from the final structures

of the $O_o(E26^-/D34^P)$ and $I_f(E26^-/D34^P)$ runs calculated for the last 2.7μs trajectory of the $O_o(E26^-/D34^P)$ MD run were 1.24 ± 0.14 and 1.21 ± 0.08 Å, respectively. Corresponding values calculated for the last 0.5μs trajectory of the $I_f(E26^-/D34^P)$ MD run were 1.48 ± 0.16 and 1.23 ± 0.26 Å, respectively. Thus the two simulations converged to similar states.

## Data availability

Data supporting the findings of this manuscript are available from the corresponding authors upon reasonable request. Coordinates of the MdfA–Fab complex have been deposited in the Protein Data Bank under the accession number 6GV1 (https://doi.org/10.2210/pdb6GV1/pdb).

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

## Acknowledgements

We would like to thank Vincent Oleric and Takashi Tomizaki of the Swiss Light Source (SLS, Villingen) for assistance in data collection and Toshiya Senda for helpful discussions and support. This work was supported by the Bundesministerium für Bildung und Forschung (BMBF) program ZIK HALOmem (FKZ 03Z2HN21 to M.T.), by the European Regional Development Fund ERDF (1241090001 to M.T.S.), and in part by the Platform Project for Supporting in Drug Discovery and Life Science Research (Platform for Drug Discovery, Informatics and Structural Life Science (PDIS) and Basis for Supporting Innovative Drug Discovery and Life Science Research (BINDS)) from the Japan Agency for Medical Research and Development (AMED) under Grant number JP18am0101107 (to T.T.), JP18am0101079 (to S.I. and N.N), and JP18am0101071 (to M.T), the ERATO Human Receptor Crystallography Project of the Japan Science and Technology Agency (JST) (to S.I.), by the Strategic Basic Research Program, JST (to S.I. and N.N), by the Targeted Proteins Research Program of the Ministry of Education, Culture, Sports, Science and Technology (MEXT) of Japan (to S.I.), and by Grants-in-Aids for Scientific Research from the MEXT (No. 22570114 to N.N.). Crystallographic data were collected at the SLS with support from the European Community's 7th Framework Programme (FP7/2007–2013) under BioStruct-X (grant agreement No. 283570, project ID: BioStructx_5450 to M.T.).

## Author contributions

K.N., F.J. and M.T. purified and crystallized MdfA and collected diffraction data. Fab YN1074 was produced by Y.N.-N., K.L., Y.H. and N.N. using the antibody production platform for membrane proteins developed by S.I. and N.N. The structure was solved by K.N. and C.P. and analyzed together with M.T.S. and M.T. Reconstitution experiments were performed by N.J., T.M., H.O. and M.T. MD simulations were carried out by T.T. The project was conceived and designed by M.T. and supervised together with M.T.S. All authors participated in analysis and discussion of the results and contributed to the preparation of the manuscript.

## Additional information

**Competing interests:** The authors declare no competing interests.

