## [Peer Review File · Nature Communications]

Reviewers' comments:

Reviewer #1 (Remarks to the Author):

The study by Nagarathinam et al., reports the outward facing crystal structure of the E. coli multi drug antiporter MdfA. The structure was determined using a FAB based crystallization chaperone and determined to 3.4 Angstrom resolution and the crystallographic data look good. The main point of this study is that the authors have trapped an outward facing state of MdfA. Previously structures of the inward facing state have been reported in complex with the drug chloramphenicol, deoxycholate and LDAO (Cell Research, 2015) and apparently also with acetylcholine and reserpine (Biophys Rep 2016), although these structures are not in the PDB? As such the authors are in a strong position to compare their outward open model to the inward facing, substrate bound states reported. In doing this analysis the authors identify that in their structure TM5 appears to adopt a conformationally strained position within the transporter, which coincidentally occurs within the region of the antiporter motif (APXXGP) that contains two prolines. The study highlights that the strained conformation of TM5 is the result of interactions between TM1 and TM4 that form in the outward facing state. The authors propose a transport model wherein protonation of the residues holding TM1 and 4 together result in relaxation of the N-terminal bundle, allowing TM5 to relax into an ideal geometry and in so doing switch the transporter to the inward facing state ready to receive additional drugs for export. In so doing the model seeks to explain how the inwardly directed proton gradient is able to drive drug efflux.

The major issue with the current study, and one that for me precludes publication, is that the authors provide no evidence for their hypothesis concerning TM5 in the transport mechanism. I agree with them that the structural comparison is suggestive of a mechanism that involves the antiporter motif and the structural rearrangement of TM5. However, the evidence provided is only suggestive. I agree that the genetic rescue mutants, which are used to support this model, are also consistent with the mechanism. But they do not, in themselves provide insight into the detailed molecular mechanism being proposed in this study.

The authors need to consider what evidence they can provide to support their claim that TM5 operates like a 'Torsion-spring' in driving the outward to inward transition.

I think that given the technical difficulties in providing experimental evidence for this model to only feasible option is molecular dynamics. Here the authors can choose from numerous studies that have looked at similar systems, indeed, they also reference a study that looked at the fucose symporter, FucP (Biophysical J.109, 542–551 (2015)). Such analysis would enable the authors to test their hypotheses concerning the interactions between Tyr127 and Glu26 and rationalize the rescue mutants. It would also enable them to study the energy landscape of the system and measure the torque generated when moving to the outward facing state. Indeed, their statement at the end of page 7, line 207 concerning the energy landscape of MdfA is currently unfounded. Ultimately it should be possible to test the current model using MD or even steered MD. Currently the present study presents plausible ideas supported by a good crystal structure and consistent genetic data, but presents no experimental or simulation data to support or rationalize these suggestions.

Also absent in this study is any mention of repeat swapping or indeed, how well a repeat swapped model of the inward and outward facing states can be modeled using this method. The authors use a rather out of date model for MFS transport that considers the N- and C-terminal halves as rigid bodies. Recent evidence from the glucose transporters, peptide transporters and lacY now suggest that individual helices within these domains are mobile and operate to switch the conformation. Can the authors discuss whether they think MdfA operates using a similar mechanism and whether they see a reciprocal movement in TM11, the symmetry related helix to TM5.

Reviewer #2 (Remarks to the Author):

Tanabe and co-workers performed an impressive crystallographic work, and solve the first structure of MdfA in an outward-facing state. The structural comparison with a previously-solved crystal structure of MdfA in the inward-facing state, allowed the authors to propose an interesting alternating-access mechanism based on a "spring-like" torsion of TM5. This mechanism is a variation of the "rocker-switch" mechanism first described for LacY, another prokaryotic MFS transporter, and Tanabe et al. propose that the TM5 torsion might be a general mechanism for the MFS antiporters.

The crystallographic studies presented by the Tanabe group in this manuscript are sound and a "tour-de-force", including the development of conformationally-selective crystallization chaperones (Fabs), and provide an important structural snapshot of the MFS-antiporters' working cycle.

However, I have some major concerns that should be addressed to strengthen the proposed model:

1- The current manuscript lacks direct functional evidence/support for the proposed TM5-torsion model. The only functional evidence for the proposed mechanism comes from "in-vivo" studies from the Bibi's laboratory. The Tanabe laboratory has the expertise to reconstitute the transporter in liposomes, since they generated Fabs against reconstituted protein, and transport assays to compare the function of the WT and mutant transporters are called upon. First, it is unclear if the purified crystallization construct is functional when reconstituted in synthetic liposomes. Second, if the construct is functional, the authors should directly test their hypothesis that an alternating H-bond between Y127 and E26 is key to isomerize the transporter between the outward- and inward-facing states. A particularly interesting position to explore is Y127, since it is hypothesized that it could form H-bonds with other positions than E26. These experiments will help clarify if a Y127-E26 H-bond is essential and the need for a "strong" H-bond to achieve translocation.

2-The authors also describe conformational changes in L41-V54, a region to which they attribute some potential role in re-locating the transporter from inward- to outward-facing through "lateral propagation" and alteration of R112 H-bonding. Once again mutations trying to rationalize these ideas would be very reassuring. The authors mentioned some mutagenesis reported in the literature around this basic residue, but don't make any attempts to interpret specific mutations in the context of their hypothesis; also the region R281-V284 shows conformational changes compared to the inward-facing state, but the authors exclude it from being mechanistically relevant on the basis that it showed "structural variability" in the inward-facing state with different ligands and it might be inherently flexible (figure 2, legend). Is there any difference in the b-factors of this region of the protein compared to others? In the 4ZOW structure the b-factors of R281-V284 are slightly higher than the majority of the protein. How about in the present structure? Could it be that the structural changes seen in this region with different ligands reflect a tendency of such ligands to isomerize the transporter into the outward-facing state, and that the region is relevant for the translocation mechanism? One more time, functional studies of mutants in this region compared to the wild-type could help the authors to shed light on this issue, or at least a more in-depth discussion in the body of the text on what the basis are to exclude this region as relevant.

3- More than half of the method section is dedicated to describe MD simulations that bring very little to the proposed mechanism or the transport mechanism in general. Moreover, I couldn't find any mention of the simulations or table S2 in the text. Do the simulations show any characteristic difference in dynamics of the regions highlighted in figure 2. For instance, is R281-V284 more or less dynamic than other regions? Could MD simulations be used to explore more insightful questions, like dynamics of TM5 or the effect of substrate binding on those dynamics in the outward-facing state? If the MD simulations are to be included in the manuscript, I strongly

suggest describing and discussing them in the body of the text and stating clearly what insights they bring to this work.

In summary, the sound crystallographic study by Tanabe and co-workers contrast with the lack of functional data, and of insights from MD simulations.

Reviewer #3 (Remarks to the Author):

Nagarathinam et al. report a crystal structure of the MDR antiporter MdfA, co-crystallized with a conformation-specific antibody fragment. The protein adopts an outward-open state, and thus complements the structure of an inward-open state reported previously by Heng et al. The study includes no biochemical, functional or cell-biological data. A molecular dynamics simulation was carried out to assess the validity of the structure in the absence of the Fab fragment.

While new structural information is always of interest, the limited scope of this study makes this contribution incremental and largely speculative. Comparison of the two structures of MdfA confirms what has been observed for other transporters in the same family (MFS) – namely that their alternating-access mechanism involves a process whereby the N- and C- terminal domains rotate around the substrate-binding site, which remains approximately in place and thus becomes exposed to either side of the membrane. Early in the manuscript the authors state that “Despite progress in structural determinations [...] for uniporter and symporter MFS transporters, the mechanisms that govern switching in antiporters remain elusive.” That this mechanism was “elusive” was not apparent to me. Why would MFS antiporters have a conformational mechanism drastically different from that of symporters or uniporters? I would expect the overall mechanism to be highly similar. What determines whether a transporter functions as a symporter or an antiporter is what substrate occupancy states permit or prohibit the alternating-access transition; to understand that functional specificity is indeed a key question – but this question cannot be addressed simply by visual inspection of one or two crystal structures.

The authors discuss more subtle differences between the two MdfA structures – and particularly highlight those observed in helix TM5 (Fig. 3a-c) and neighboring regions. These changes form the basis for the mechanistic interpretation of the structure. Specifically, the authors speculate that in the outward-open state this helix is “tensed” and that it becomes “relaxed” in the inward-open state – and further infer that this process of “relaxation” is akin to a “torsion-spring” that is “responsible for flipping between the outward and inward open states”. This notion is problematic on multiple levels. First, it could be reasonably argued, based on Figure S2, and on what has been observed in structures of other transporters (e.g. the sodium-calcium antiporter), that the subtle differences observed in TM5 reflect that in the current structure no ligand is bound, in contrast to the earlier study of the inward-facing state. Without a structure of the outward-facing state in the bound form or a structure of the inward-facing state in the apo form, it is entirely unclear what changes in the structure reflect drug or H⁺ binding and what changes reflect the inward-to-outward transition. Second, the notion that the outward-facing state is somehow strained owing to the conformation of TM5, like a loaded spring, is very questionable, in my opinion. Clearly, the protein is observed in a free-energy minimum – or else the authors would not have been able to crystallize it and resolve its atomic structure. A structure at a free-energy minimum is not strained, but nonetheless will interconvert with other states that are also free-energy minima – as dictated by the Boltzmann distribution. Mechanical analogies are popular in other fields but it is far from evident they are applicable to the mechanism of secondary-active transporters. In summary, I regret to say that I find that the authors’ major claim, as stated in e.g. the title of the manuscript, is not convincingly supported by the data presented.

Other issues:

1) The authors report a molecular dynamics simulation of the protein in a phospholipid membrane,

with which they “evaluate the stability of the outward-open conformation of MdfA in the absence of Fab”. The results of the simulation are presented in terms of time-averaged RMSD values (Table S2). Given that the purpose of the simulation is to assess the stability of the structure, the authors should report plots of the time-series of each of these RMSD values.

2) Figure 5. The model described here very clearly implies that the structure presented in this study reflects a state that can transition to the inward-facing state, i.e. it is bound to H⁺. However, this interpretation is inconsistent with the text, e.g. lines 175-177. Either the figure is altered, or the text revised. It also appears that the simulation is of a deprotonated state. However, the pKa estimated for Glu26 and Asp34 is said to be around 7 – implying the protonation probability at pH 6.4 is greater than 50%.

3) Figure 2. It is unclear what RMSD stands for in this case. I believe the authors quantify the differences between the two MdfA structures in terms of the distance between corresponding C-alpha atoms, after fitting one structure onto the other (domain by domain). Since only two structures are compared, and only one atom is considered per residue, this quantity is not a root-mean-squared distance. It is just a distance.

Point-to point response to reviewers

Reviewer #1: The study by Nagarathinam et al., reports the outward facing crystal structure of the E. coli multi drug antiporter MdfA. The structure was determined using a FAB based crystallization chaperone and determined to 3.4 Angstrom resolution and the crystallographic data look good. The main point of this study is that the authors have trapped an outward facing state of MdfA. Previously structures of the inward facing state have been reported in complex with the drug chloramphenicol, deoxycholate and LDAO (Cell Research, 2015) and apparently also with acetylcholine and reserpine (Biophys Rep 2016), although these structures are not in the PDB? As such the authors are in a strong position to compare their outward open model to the inward facing, substrate bound states reported. In doing this analysis the authors identify that in their structure TM5 appears to adopt a conformationally strained position within the transporter, which coincidentally occurs within the region of the antiporter motif (APXXGP) that contains two prolines. The study highlights that the strained conformation of TM5 is the result of interactions between TM1 and TM4 that form in the outward facing state. The authors propose a transport model wherein protonation of the residues holding TM1 and 4 together result in relaxation of the N-terminal bundle, allowing TM5 to relax into an ideal geometry and in so doing switch the transporter to the inward facing state ready to receive additional drugs for export. In so doing the model seeks to explain how the inwardly directed proton gradient is able to drive drug efflux.

The major issue with the current study, and one that for me precludes publication, is that the authors provide no evidence for their hypothesis concerning TM5 in the transport mechanism. I agree with them that the structural comparison is suggestive of a mechanism that involves the antiporter motif and the structural rearrangement of TM5. However, the evidence provided is only suggestive. I agree that the genetic rescue mutants, which are used to support this model, are also consistent with the mechanism. But they do not, in themselves provide insight into the detailed molecular mechanism being proposed in this study.

The authors need to consider what evidence they can provide to support their claim that TM5 operates like a 'Torsion-spring' in driving the outward to inward transition.

I think that given the technical difficulties in providing experimental evidence for this model to only feasible option is molecular dynamics. Here the authors can choose from numerous studies that have looked at similar systems, indeed, they also reference a study that looked at the fucose symporter, FucP (Biophysical J.109, 542–551 (2015)). Such analysis would enable the authors to test their hypotheses concerning the interactions between Tyr127 and Glu26 and rationalize the rescue mutants. It would also enable them to study the energy landscape of the system and measure the torque generated when moving to the outward facing state. Indeed, their statement at the end of page 7, line 207 concerning the energy landscape of MdfA is currently unfounded. Ultimately it should be possible to test the current model using MD or even steered MD. Currently the present study presents plausible ideas supported by a good crystal structure and consistent genetic data, but presents no experimental or simulation data to support or rationalize these suggestions.

Following the advice of the reviewer, we have performed comprehensive MD simulations with all permutations of protonated/deprotonated Glu26^{TM1}/Asp34^{TM1}. Starting from our outward open (O_o) structure, we show that the largest changes occur upon protonation of Asp34^{TM1}, which leads to an occluded state in which the acidic side chain becomes enclosed in an internal cavity that is also found in the I_f conformation. In this occluded state, TM5 remains twisted and the Glu26^{TM1}- Tyr127^{TM4} hydrogen bond is largely maintained. Initiating the simulations from the inward facing (I_f) state following removal of the ligand promotes TM5 twisting and a close approach of Tyr127^{TM4} and Glu26^{TM1}. We therefore conclude that, starting from the O_o state, TM5 untwisting occurs following formation of the occluded state, either upon achieving the inward open I_o state or after ligand binding to adopt the I_f state.

Also absent in this study is any mention of repeat swapping or indeed, how well a repeat swapped model of the inward and outward facing states can be modeled using this method. The authors use a rather out of date model for MFS transport that considers the N- and C-terminal halves as rigid bodies. Recent evidence from the glucose transporters, peptide transporters and lacY now suggest that individual helices within these domains are mobile and operate to switch the conformation. Can the authors discuss whether they think MdfA operates using a similar mechanism and whether they see a reciprocal movement in TM11, the symmetry related helix to TM5.

We realize in retrospect that describing the N- and C-domains moving as rigid bodies (which is – with the exception of TM5 –indeed the case when comparing O_o and I_f structures) is misleading. We now point this out explicitly in the discussion (lines 264-266: “Recent structure determinations of other transporters⁵⁻⁷ indicate that individual helices within each domain can exhibit significant variability upon conformation switching”). We do not see any reciprocal movement in TM11 in the crystal structures, but we point out the situation postulated for FucP might be considered as being analogous (lines 302-305: “Interestingly, C-terminal domain TM11 is the counterpart of TM5 in the (inverted topology) N-terminal domain²³, reflecting the pseudosymmetry of the two domains, so that the antiporter MdfA and symporter FucP might be thought of as examples of repeat swapping to yield similar transport mechanisms”).

Reviewer #2: Tanabe and co-workers performed an impressive crystallographic work, and solve the first structure of MdfA in an outward-facing state. The structural comparison with a previously-solved crystal structure of MdfA in the inward-facing state, allowed the authors to propose an interesting alternating-access mechanism based on a “spring-like” torsion of TM5. This mechanism is a variation of the “rocker-switch” mechanism first described for LacY, another prokaryotic MFS transporter, and Tanabe et al. propose that the TM5 torsion might be a general mechanism for the MFS antiporters.

The crystallographic studies presented by the Tanabe group in this manuscript are sound and a “tour-de-force”, including the development of conformationally-selective crystallization chaperones (Fabs), and provide an important structural snapshot of the MFS-antiporters’ working cycle.

However, I have some major concerns that should be addressed to strengthen the proposed model:

1- The current manuscript lacks direct functional evidence/support for the proposed TM5-torsion model. The only functional evidence for the proposed mechanism comes from “in-vivo” studies from the Bibi’s laboratory. The Tanabe laboratory has the expertise to reconstitute the transporter in liposomes, since they generated Fabs against reconstituted protein, and transport assays to compare the function of the WT and mutant transporters are called upon. First, it is unclear if the purified crystallization construct is functional when reconstituted in synthetic liposomes. Second, if the construct is functional, the authors should directly test their hypothesis that an alternating H-bond between Y127 and E26 is key to isomerize the transporter between the outward- and inward-facing states. A particularly interesting position to explore is Y127, since it is hypothesized that it could form H-bonds with other positions than E26. These experiments will help clarify if a Y127-E26 H-bond is essential and the need for a “strong” H-bond to achieve translocation.

We have established the reconstitution of purified MdfA in liposomes, allowing us to measure the chloramphenicol transport activity of the constructs used for crystallization in the presence and absence of a pH gradient. Purified reconstituted wild-type MdfA was able to transport 50 pmol chloramphenicol (per mg protein per minute), which compares favorably with the 3 pmol / mg / min determined using crude membrane preparations. Whereas transport proved unaffected by mutation of Glu26^{TM1} to Gln, suggesting that the charge state of this residue is not crucial for chloramphenicol transport, the variants Tyr127^{TM4}Phe, Met146^{TM5}Ala showed significant reductions in chloramphenicol transport in the presence of a pH gradient (Figure 4). As expected, no transport was observed in the absence of ΔpH. These data provide evidence that the Tyr127^{TM4} hydroxyl moiety and the Met146^{TM5} side chain are requirements for activity, but that a negatively charged Glu26^{TM1} is not necessary for

chloramphenicol transport. This is in line with our interpretation that the kink in TM5 is influenced by mutually exclusive interactions between Tyr127^{TM4} and Met146^{TM5} / Glu26^{TM1}.

2-The authors also describe conformational changes in L41-V54, a region to which they attribute some potential role in re-locating the transporter from inward- to outward-facing through “lateral propagation” and alteration of R112 H-bonding. Once again mutations trying to rationalize these ideas would be very re-assuring. The authors mentioned some mutagenesis reported in the literature around this basic residue, but don’t make any attempts to interpret specific mutations in the context of their hypothesis; also the region R281-V284 shows conformational changes compared to the inward-facing state, but the authors exclude it from being mechanistically relevant on the basis that it showed “structural variability” in the inward-facing state with different ligands and it might be inherently flexible (figure 2, legend). Is there any difference in the b-factors of this region of the protein compared to others? In the 4ZOW structure the b-factors of R281-V284 are slightly higher than the majority of the protein. How about in the present structure? Could it be that the structural changes seen in this region with different ligands reflect a tendency of such ligands to isomerize the transporter into the outward-facing state, and that the region is relevant for the translocation mechanism? One more time, functional studies of mutants in this region compared to the wild-type could help the authors to shed light on this issue, or at least a more in-depth discussion in the body of the text on what the basis are to exclude this region as relevant.

Using the aforementioned reconstituted system, we have also probed the importance of the hydrophobic core adjacent to Arg112^{TM4} on transport through mutation of Trp170^{TM6}, with the variant Trp170^{TM6}Ala demonstrating a loss of pH-dependent chloramphenicol transport. We accept that there is insufficient information to assign a “natural structural variability” to the region Arg281^{TM8}-Val284^{TM9}, and point out instead that, as the contact site for the TM5 N-terminal residues in the O_o state (see Figure 2), the “cytoplasmic loop of the C-terminal domain rearranges to accommodate closure of the cytoplasmic entrance” (lines 239:240). In general, B-factors for the loop regions are higher than for the helical parts; taking into account the moderate resolution of the present study, we do not discuss this aspect (in any case, dynamic solid state NMR studies reveal little or no correlation between X-ray crystallographic temperature factors and protein dynamics [Reichert D, Zinkevich T, Saalwächter K & Krushelnitsky A (2012), J. Biomol. Structure and Dynamics, 30, 617-627]).

3- More than half of the method section is dedicated to describe MD simulations that bring very little to the proposed mechanism or the transport mechanism in general. Moreover, I couldn't find any mention of the simulations or table S2 in the text. Do the simulations show any characteristic difference in dynamics of the regions highlighted in figure 2. For instance, is R281-V284 more or less dynamic than other regions? Could MD simulations be used to explore more insightful questions, like dynamics of TM5 or the effect of substrate binding on those dynamics in the outward-facing state? If the MD simulations are to be included in the manuscript, I strongly suggest describing and discussing them in the body of the text and stating clearly what insights they bring to this work.

In summary, the sound crystallographic study by Tanabe and co-workers contrast with the lack of functional data, and of insights from MD simulations.

As detailed above in our reply to reviewer #1, we have carried out extensive MD simulations that are described and discussed in the main body of the text.

Reviewer #3: Nagarathinam et al. report a crystal structure of the MDR antiporter MdfA, co-crystallized with a conformation-specific antibody fragment. The protein adopts an outward-open state, and thus complements the structure of an inward-open state reported previously by Heng et al. The study includes no biochemical, functional or cell-biological data. A molecular dynamics simulation was carried out to assess the validity of the structure in the absence of the Fab fragment.

While new structural information is always of interest, the limited scope of this study makes this contribution incremental and largely speculative. Comparison of the two structures of MdfA confirms what has been observed for other transporters in the same family (MFS) – namely that their alternating-access mechanism involves a process whereby the N- and C- terminal domains rotate around the substrate-binding site, which remains approximately in place and thus becomes exposed to either side of the membrane. Early in the manuscript the authors state that “Despite progress in structural determinations [...] for uniporter and symporter MFS transporters, the mechanisms that govern switching in antiporters remain elusive.” That this mechanism was “elusive” was not apparent to me. Why would MFS antiporters have a conformational mechanism drastically different from that of symporters or uniporters? I would expect the overall mechanism to be highly similar. What determines whether a transporter functions as a symporter or an antiporter is what substrate occupancy states permit or prohibit the alternating-access transition; to understand that functional specificity is indeed a key question – but this question cannot be addressed simply by visual inspection of one or two crystal structures.

We accept that the use of the phrase “elusive” was misleading, and have replaced the corresponding sentence with “Despite progress in structural determinations of these states for uniporter and symporter MFS transporters, few such data are available for antiporters.” (lines 61:63).

The authors discuss more subtle differences between the two MdfA structures – and particularly highlight those observed in helix TM5 (Fig. 3a-c) and neighboring regions. These changes form the basis for the mechanistic interpretation of the structure. Specifically, the authors speculate that in the outward-open state this helix is “tensed” and that it becomes “relaxed” in the inward-open state – and further infer that this process of “relaxation” is akin to a “torsion-spring” that is “responsible for flipping between the outward and inward open states”. This notion is problematic on multiple levels. First, it could be reasonably argued, based on Figure S2, and on what has been observed in structures of other transporters (e.g. the sodium-calcium antiporter), that the subtle differences observed in TM5 reflect that in the current structure no ligand is bound, in contrast to the earlier study of the inward-facing state. Without a structure of the outward-facing state in the bound form or a structure of the inward-facing state in the apo form, it is entirely unclear what changes in the structure reflect drug or H⁺ binding and what changes reflect the inward-to-outward transition. Second, the notion that the outward-facing state is somehow strained owing to the conformation of TM5, like a loaded spring, is very questionable, in my opinion. Clearly, the protein is observed in a free-energy minimum – or else the authors would not have been able to crystallize it and resolve its atomic structure. A structure at a free-energy minimum is not strained, but nonetheless will interconvert with other states that are also free-energy minima – as dictated by the Boltzmann distribution. Mechanical analogies are popular in other fields but it is far from evident they are applicable to the mechanism of secondary-active transporters. In summary, I regret to say that I find that the authors’ major claim, as stated in e.g. the title of the manuscript, is not convincingly supported by the data presented.

We agree absolutely with the reviewer that additional structural information will be necessary to provide a robust model for drug/proton antiport. In particular, experimental details on the occluded states identified here using MD simulations will be required to dissect individual steps in the transport cycle. Nevertheless, we feel that the combined structural, biochemical and computational data presented here provide a sound basis for such understanding.

Other issues:

1) The authors report a molecular dynamics simulation of the protein in a phospholipid membrane, with which they “evaluate the stability of the outward-open conformation of MdfA in the absence of Fab”. The results of the simulation are presented in terms of time-averaged RMSD values (Table S2). Given that the purpose of the simulation is to assess the stability of the structure, the authors should report plots of the time-series of each of these RMSD values.

A plot showing the time evolution of the RMSD values shows that these fluctuate around the average values (depicted here for the reviewer), so that we feel that the data shown in Table S2 are sufficient to explain the stability of the protein.

2) Figure 5. The model described here very clearly implies that the structure presented in this study reflects a state that can transition to the inward-facing state, i.e. it is bound to H⁺. However, this interpretation is inconsistent with the text, e.g. lines 175-177. Either the figure is altered, or the text revised. It also appears that the simulation is of a deprotonated state. However, the pK_a estimated for Glu26 and Asp34 is said to be around 7 – implying the protonation probability at pH 6.4 is greater than 50%.

We thank the reviewer for pointing out this inconsistency, and now utilize clearer expressions in the text. As discussed above in our replies to the other two reviewers, we have now expanded the MD simulations considerably, addressing all possible permutations for the protonation of Glu26^{TM1}/Asp34^{TM1}.

3) Figure 2. It is unclear what RMSD stands for in this case. I believe the authors quantify the differences between the two MdfA structures in terms of the distance between corresponding C-alpha atoms, after fitting one structure onto the other (domain by domain). Since only two structures are compared, and only one atom is considered per residue, this quantity is not a root-mean-squared distance. It is just a distance.

The reviewer is of course correct in saying that the plot formally represents a distance. Such representations of the RMSD between two structures (a single value overall) on a per-residue basis are however routinely used to identify regions of variability within a sequence. In our opinion, labelling the axis "distance" would only serve to foster confusion.

Reviewers' comments:

Reviewer #1 (Remarks to the Author):

The authors have addressed the main concerns I had regarding the interpretation of the structural changes in TM5. The manuscript is now more descriptive and the references to 'torsion-spring' have been removed.

The authors support their functional data with molecular simulation analysis after protonating/deprotonating the two conserved acidic residues in the binding site (E26, D34). These results demonstrate that D34 is the key protonatable residue (which was known previously) but add mechanistic insight into the role of this side chain in switching the conformation from outward to inward open. There is a missing control in these simulations, which would be the E26Q variant. Does this behave in the same way as E26p.

I was a little confused about the role the authors ascribe to Glu26. The data suggests that charge state of E26 is of little importance for chloramphenicol transport. But do the authors think this is the case for all substrates of MdfA? This is currently unclear. In the discussion the authors refer in line 250 'two important acidic residues Asp34 and Glu26.'. But, have they not shown that E26 is not important? Or that just the charged state is not important. Please clarify.

Overall this is an important contribution to the drug transport field, as it shows a novel state of a major bacterial multi drug transporter and illuminates important aspects of the transport mechanism.

Reviewer #2 (Remarks to the Author):

Tanabe and co-workers have greatly improved the manuscript by establishing a transport assay that allowed them to challenge the proposed mechanism by mutagenesis. Their functional results with mutant Y127F, as well as E126Q and M146A strengthen the proposed mechanism involving exclusive interactions of the aromatic residue with the other two amino acids.

They expanded, presented and discussed their MD simulations in a more meaningful way contributing to the overall improvement of the manuscript and the conclusions.

The authors have addressed my concerns and I don't have further comments.

Reviewer #3 (Remarks to the Author):

Nagarathinam et al. have revised their manuscript significantly – specifically they have added functional data as well as new/extended molecular dynamics simulations. Unfortunately, the new data is not only not presented in sufficient amount of detail, but it also appears to reveal rather fundamental problems.

The functional data originates in chloramphenicol "transport" assays in a liposome system – with MdfA reconstituted. The authors set up the experiment such that the luminal pH is 6 and the external pH is either 6 or 7.5. Chloramphenicol is on the outside but not in the lumen – i.e. effectively the so-called "infinite gradient" condition. On page 7, the authors state that "As expected, no transport was observed in the absence of Δ pH". Indeed, the measurement in this condition, i.e. the first bar in the plot in Fig. 4, becomes the "yard stick" to evaluate the importance or lack thereof of a range of putatively notable residues.

The problem is that chloramphenicol uptake should have been observed even when Δ pH = 0, if MdfA is indeed an H⁺-coupled antiporter, as it is claimed. What is expected under this condition is downhill uptake of chloramphenicol in exchange for uphill efflux of protons. Naturally, the rate of

chloramphenicol uptake would be slower in this condition than when H⁺ efflux is also downhill, i.e. when the external pH is 7.5 - but uptake would nevertheless occur. It is therefore questionable whether the "inactive" reference in Fig. 4 is in fact so. If it is, as the authors claim, then MdfA does not behave like a H⁺-coupled antiporter in these assays. If it is not, then it is also unclear how to interpret the impact that the various mutations supposedly have - e.g. it might appear as if none actually abrogate uptake.

Clearly, the authors need to expand this functional analysis and provide additional controls, e.g. uptake measurements for protein-free liposomes with identical conditions and procedures. The fact that chloramphenicol is lipid soluble makes this control particularly important. For the same reason, it is essential that the authors demonstrate, for the WT protein, actual transport (and not merely downhill translocation along an infinite gradient), i.e. active uptake of chloramphenicol into chloramphenicol-loaded liposomes driven by downhill H⁺ efflux. Without having established this foundation, it is simply not possible to evaluate the authors' conclusions in regard to the mutants. Incidentally, the authors should provide a representative sample of the data that underlies the bar chart in Fig. 4, e.g. time-dependent radioactivity count in different conditions, etc. In summary, as it stands, the functional data in this paper, following the authors' interpretation, is clearly inconsistent with the notion that MdfA is a H⁺-driven antiport system - and so I cannot recommend publication.

The newly provided simulation data is similarly self-contradictory. The primary claim from these simulations is that protonation of Asp34 leads to some kind of occlusion in simulations initiated in the outward-open and (putative) inward-state. The authors state that the end-point of these two simulations are "similar", but do not quantify the actual similarity between these hypothetically occluded states or the time-course of the simulations towards that point. I would want to see RMSD time-series using each of these four states as the reference to be convinced that both simulations are converging towards a state that is "similar", relative to the starting point (in the actual submission, not the rebuttal).

A more important but related concern is the time-scale and reversibility of the proposed changes. The time-series included in the rebuttal shows structural changes that are extremely fast (~10 ns) and irreversible. It is not clear what simulation this data corresponds to - possibly that of MdfA deprotonated at Asp34 and Glu26. If so, and if the simulations for the protonated form of Asp34 shows changes that are similarly fast and/or similarly irreversible, this data would severely undermine the claim that these motions are mechanistically significant. To be clear, I completely agree that protonation of a H⁺-coupled antiporter should foster the formation of an occluded state, which would be not accessible in the deprotonated state - that is the essence of H⁺ coupling in secondary-active transport. However, this "occlusion" process must be fully reversible, i.e. the open and occluded conformations must coexist in equilibrium. Otherwise, no transport would occur, clearly. For example, the inward-facing occluded state would not be able to unload the bound H⁺ and replace it with a substrate molecule unless it is able to adopt an open state. An analogous argument could be made for the outward facing state, evidently. And so, inward and outward open states must appear as (meta)stable states with Asp34 protonated, if this residue is in fact the H⁺ carrier in the mechanism of the protein. I deduce from the limited information provided on page 9 that what the authors observed are irreversible, fast structural changes, which, as mentioned, undermine the claim that these motions reflect the actual conformational mechanism of the protein, as opposed to changes driven by other factors (suboptimal preparation, removal of Fab fragment, etc). Evidently, this kind of collapse would not occur for the deprotonated form, as dehydration of Asp34 without a compensating interaction would be energetically very costly. In conclusion, as presented, also the newly provided simulation data appears to contradict fundamental expectations for a H⁺-coupled antiport system.

Point-by-point response to reviewers

Reviewer #1: The authors have addressed the main concerns I had regarding the interpretation of the structural changes in TM5. The manuscript is now more descriptive and the references to 'torsion-spring' have been removed.

The authors support their functional data with molecular simulation analysis after protonating/deprotonating the two conserved acidic residues in the binding site (E26, D34). These results demonstrate that D34 is the key protonatable residue (which was known previously) but add mechanistic insight into the role of this side chain in switching the conformation from outward to inward open. There is a missing control in these simulations, which would be the E26Q variant. Does this behave in the same way as E26p.

We generated models of the E26Q variant using the crystal structures of the O_o and I_f states [hereafter denoted O_o(E26Q) and I_f(E26Q), respectively] and performed MD simulations for 1 μs with and without D34 protonation. The figure below shows the conformational distribution in the MD simulations as a function of d_1 and d_2 defined in Figure 5.

The results of the MD runs starting from the O_o(E26Q/D34⁻) and I_f(E26Q/D34p) models were similar to those from O_o(E26p/D34⁻) and I_f(E26p/D34p), respectively, as expected. In contrast to the results for O_o(E26p/D34p), however, no transition to the occluded state was observed in the O_o(E26Q/D34p) run. In this case, the N-lobe of the protein rotated around the axis connecting the centres of the N- and C-lobes to move TM5 inward, collapsing the cavity between the lobes. The rotation axis is different (almost perpendicular) to that for the transition to the occluded state. This rotation was also observed in the MD runs O_o(E26Q/D34⁻) and I_f(E26Q/D34p) as well as O_o(E26p/D34⁻) and I_f(E26p/D34p). Finally, a transition was observed for the I_f(E26Q/D34⁻) model to a state similar to (but distinct from) the occluded state, whereas the structure of I_f(E26p/D34⁻) changed little during the simulation.

Although the MD runs showed qualitative agreement, the behaviours of the E26Q variant were not exactly the same as those of the wild-type E26p state. We assume that the carboxamide group is an imperfect mimic for a protonated carboxylate. In particular, the extra proton in E26Q could subtly influence interactions with Y127 and Y30, which could in turn affect the structure of the O_o state. Analysing the behaviour of this variant in detail, however, would require experimental structures of the E26Q variant, which are beyond the scope of this paper. We would therefore rather not discuss these results in the present manuscript.

I was a little confused about the role the authors ascribe to Glu26. The data suggests that charge state of E26 is of little importance for chloramphenicol transport. But do the authors think this is the case for all substrates of MdfA? This is currently unclear. In the discussion the authors refer in line 250 'two important acidic residues Asp34 and Glu26.'. But, have they not shown that E26 is not important? Or that just the charged state is not important. Please clarify.

The role of E26 is still not fully clear. Our present study suggests the protonation of E26 alone does not contribute to the conformational transition between the O_o and occluded states, which should be equivalent for all substrates. On the other hand, previous *in vivo* studies [Adler et al., 2004 / #16; Sigal et al., 2009 / #11] have shown E26 to be critical for the transport of cationic substrates, but not so important for neutral substrates including chloramphenicol. We have therefore modified the corresponding sentence “we performed molecular dynamics simulations involving different protonation states of the two acidic residues identified previously as being important in *in vivo* studies^{11,16,21}, Asp34^{TM1} and Glu26^{TM1}” (page 11).

We have also now added the proviso “It should be noted that, strictly speaking, the conclusions presented here apply only to chloramphenicol transport (for which structural data of the I_f form are available). Whereas we expect them to be generally valid for other neutral MdfA substrates, the situation may differ for lipophilic and cationic substrates” (page 11).

Finally, we have supplemented the sentence “TM5 continues to be twisted in this occluded state and the Glu26^{TM1}- Tyr127^{TM4} hydrogen bond remains intact” with “, although the charge state of Glu26^{TM1} does not appear to play a role in this. Nevertheless, previous *in vivo* studies have shown that Glu26^{TM1} is critical for the transport of cationic substrates^{11,16}, so that the situation may be different for cationic and lipophilic substrates, in which the “initial” I_f state assumed here might not apply” (page 11).

Overall this is an important contribution to the drug transport field, as it shows a novel state of a major bacterial multi drug transporter and illuminates important aspects of the transport mechanism.

Reviewer #2: Tanabe and co-workers have greatly improved the manuscript by establishing a transport assay that allowed them to challenge the proposed mechanism by mutagenesis. Their functional results with mutant Y127F, as well as E126Q and M146A strengthen the proposed mechanism involving exclusive interactions of the aromatic residue with the other two amino acids. They expanded, presented and discussed their MD simulations in a more meaningful way contributing to the overall improvement of the manuscript and the conclusions. The authors have addressed my concerns and I don't have further comments.

Reviewer #3: Nagarathinam et al. have revised their manuscript significantly – specifically they have added functional data as well as new/extended molecular dynamics simulations. Unfortunately, the new data is not only not presented in sufficient amount of detail, but it also appears to reveal rather fundamental problems.

The functional data originates in chloramphenicol “transport” assays in a liposome system – with MdfA reconstituted. The authors set up the experiment such that the luminal pH is 6 and the external pH is either 6 or 7.5. Chloramphenicol is on the outside but not in the lumen – i.e. effectively the so-called “infinite gradient” condition. On page 7, the authors state that “As expected, no transport was observed in the absence of Δ pH”. Indeed, the measurement in this condition, i.e. the first bar in the plot in Fig. 4, becomes the “yard stick” to evaluate the importance or lack thereof of a range of putatively notable residues.

The problem is that chloramphenicol uptake should have been observed even when Δ pH = 0, if MdfA is indeed an H⁺-coupled antiporter, as it is claimed. What is expected under this condition is downhill uptake of chloramphenicol in exchange for uphill efflux of protons. Naturally, the rate of chloramphenicol uptake would be slower in this condition than when H⁺ efflux is also downhill, i.e. when the external pH is 7.5 - but uptake would nevertheless occur. It is therefore questionable whether the “inactive” reference in Fig. 4 is in fact so. If it is, as the authors claim, then MdfA does not behave like a H⁺-coupled antiporter in these assays. If it is not, then it is also unclear how to interpret the impact that the various mutations supposedly have – e.g. it might appear as if none actually abrogate uptake.

We concur with the reviewer that the statement “As expected, no transport was observed in the absence of Δ pH” is misleading, and have replaced it with “As expected, chloramphenicol transport was low in the absence of Δ pH, arising from downhill transport due to the initial infinite substrate gradient” (page 7). The corresponding experimental evidence for downhill transport in the absence of Δ pH is described below.

Clearly, the authors need to expand this functional analysis and provide additional controls, e.g. uptake measurements for protein-free liposomes with identical conditions and procedures. The fact that chloramphenicol is lipid soluble makes this control particularly important. For the same reason, it is essential that the authors demonstrate, for the WT protein, actual transport (and not merely downhill translocation along an infinite gradient), i.e. active uptake of chloramphenicol into chloramphenicol-loaded liposomes driven by downhill H⁺ efflux. Without having established this foundation, it is simply not possible to evaluate the authors' conclusions in regard to the mutants. Incidentally, the authors should provide a representative sample of the data that underlies the bar chart in Fig. 4, e.g. time-dependent radioactivity count in different conditions, etc. In summary, as it stands, the functional data in this paper, following the authors' interpretation, is clearly inconsistent with the notion that MdfA is a H⁺-driven antiport system – and so I cannot recommend publication.

We appreciate the reviewer's healthy skepticism, and have carried out the suggested controls (Figure 4, page 8). As can be seen in the revised Figure 4(a), chloramphenicol transport into the proteoliposomes is dependent upon the presence of MdfA and a pH gradient. Figure 4(b) shows the time-course for chloramphenicol uptake. In the absence of a pH gradient, “downhill-like” transport (with the substrate

gradient) occurs rapidly due to the small volume of the proteoliposomes¹. In the presence of a pH gradient, however, chloramphenicol uptake involves at least three phases: following a rapid initial downhill transport phase (not visible), uphill accumulation of the substrate in the liposomal lumen against the concentration gradient takes place at the expense of proton export (II). Within a few minutes, the situation is reversed due to lumen acidification, leading to chloramphenicol efflux (phase III). Crucially, collapse of the pH gradient through administration of the H⁺-ionophore CCCP results in rapid chloramphenicol efflux (downhill transport) until the luminal concentration reaches that observed in the absence of a pH gradient.

These time-dependent studies therefore demonstrate that our reconstituted system is a valid reporter of uphill chloramphenicol transport coupling with H⁺ flow, and we are grateful to the reviewer for this suggestion. To assist readers, we have added a schematic diagram of the assay (Figure 4c), while data for the MdfA variants now appear as Figure 4(d).

The newly provided simulation data is similarly self-contradictory. The primary claim from these simulations is that protonation of Asp34 leads to some kind of occlusion in simulations initiated in the outward-open and (putative) inward-state. The authors state that the end-point of these two simulations are “similar”, but do not quantify the actual similarity between these hypothetically occluded states or the time-course of the simulations towards that point. I would want to see RMSD time-series using each of these four states as the reference to be convinced that both simulations are converging towards a state that is “similar”, relative to the starting point (in the actual submission, not the rebuttal).

We have performed the relevant calculations and added them to the Materials & Methods (page 16) / Supplementary Information:

“Time evolutions of RMSDs in the O_o(E26⁻/D34p) and I_i(E26⁻/D34p) MD runs using their respective initial and final structures as reference were calculated (**Supplementary Figure 9**). The average values of the RMSDs from the final structures of the O_o(E26⁻/D34p) and I_i(E26⁻/D34p) runs calculated for the last 2.7- μ s trajectory of the O_o(E26⁻/D34p) MD run were 1.24 ± 0.14 Å and 1.21 ± 0.08 Å, respectively. Corresponding values calculated for the last 0.5- μ s trajectory of the I_i(E26⁻/D34p) MD run were 1.48 ± 0.16 Å and 1.23 ± 0.26 Å, respectively. Thus the two simulations converged to similar states.”

We would like to add that we do not expect a level of precision in the simulations that tells us whether we have one occluded state or several. Indeed, Quistgaard *et al.* (2016) (reference 7) have pointed out that inward occluded and outward occluded conformational states should be treated as different states. In the absence of additional structural data, we feel it would be misleading to describe the states reached in the simulations as being any more than “similar”.

A more important but related concern is the time-scale and reversibility of the proposed changes. The time-series included in the rebuttal shows structural changes that are extremely fast (~10 ns) and irreversible. It is not clear what simulation this data corresponds to – possibly that of MdfA deprotonated at Asp34 and Glu26. If so, and if the simulations for the protonated form of Asp34 shows changes that are similarly fast and/or similarly irreversible, this data would severely undermine the claim that these motions are mechanistically

¹ True downhill transport is barely measurable under these conditions: with a Δ pH of 1.5, MdfA can establish a 30 fold chloramphenicol concentration gradient, so that the maximum amount of transported chloramphenicol in the absence of Δ pH in Figure 4(b) corresponds to 2~3 pmol/mg i.e. below the detection limit.

significant. To be clear, I completely agree that protonation of a H⁺-coupled antiporter should foster the formation of an occluded state, which would be not accessible in the deprotonated state - that is the essence of H⁺ coupling in secondary-active transport. However, this "occlusion" process must be fully reversible, i.e. the open and occluded conformations must coexist in equilibrium. Otherwise, no transport would occur, clearly. For example, the inward-facing occluded state would not be able to unload the bound H⁺ and replace it with a substrate molecule unless it is able to adopt an open state. An analogous argument could be made for the outward facing state, evidently. And so, inward and outward open states must appear as (meta)stable states with Asp34 protonated, if this residue is in fact the H⁺ carrier in the mechanism of the protein. I deduce from the limited information provided on page 9 that what the authors observed are irreversible, fast structural changes, which, as mentioned, undermine the claim that these motions reflect the actual conformational mechanism of the protein, as opposed to changes driven by other factors (suboptimal preparation, removal of Fab fragment, etc). Evidently, this kind of collapse would not occur for the deprotonated form, as dehydration of Asp34 without a compensating interaction would be energetically very costly. In conclusion, as presented, also the newly provided simulation data appears to contradict fundamental expectations for a H⁺-coupled antiport system.

We agree with the reviewer that the occluded state should coexist in equilibrium with at least one of the open states when Asp34 is protonated. From the trajectories, we calculated free energy landscapes for MdfA transitions in the E26⁻/D34p state along the d₁/d₂ coordinates. The plot for the O_o(E26⁻/D34p) MD run (**Supplementary Figure 7a**) indicates that under these circumstances, the O_o state is much less stable than the occluded state, suggesting that upon protonation of D34, the transition from the O_o state to the occluded state occurs quickly and irreversibly. In contrast, the plot for the I_f(E26⁻/D34p) MD run (**Supplementary Figure 7b**) shows that both the I_f and the occluded states are in a flat free-energy well, suggesting that these two states can co-exist and the transition between the I_f state and the occluded state is reversible. As we observed only a one-way transition from the I_f state to the occluded state in the 1-μs MD simulation, the transition must be slow, presumably due to the complex and rugged nature of the original. multi-dimensional energy surface.

These results are in agreement with the reviewer's postulate that reversibility between the I_f and the occluded states is necessary for the function. However, reversibility between the O_o and the occluded state need be not necessary, as there is no need to release H⁺ to the periplasm. Rapid closure of the opening to the periplasmic side would indeed be beneficial to the efficiency of the drug transportation. Thus, we feel that our results satisfy the requirements for the protein's functional motion.

These insights are now reflected in the manuscript:

"Analysis of the free energy landscape for this transition (**Supplementary Figure 7a**) indicates that upon protonation of Asp34^{TM1}, the O_o state is much less stable than the occluded state, suggesting that the transition occurs rapidly and is in effect irreversible" (page 9).

"In contrast to the transition from the O_o state, however, the I_f and the occluded states are in a flat free-energy well (**Supplementary Figure 7b**), suggesting that these two states can co-exist when Asp34^{TM5} is protonated and that the transition between the I_f state and the occluded state is reversible. As we observed only a one-way transition from the I_f state to the occluded state in the 1-μs MD simulation, the transition must be slow, presumably due to the complex and rugged nature of the original multi-dimensional energy surface" (page 10). The corresponding subordinate clause " , suggesting that this is a free energy minimum state when Asp34^{TM5} is protonated" has been deleted.

REVIEWERS' COMMENTS:

Reviewer #1 (Remarks to the Author):

The revised study by Nagarathinam et al. addresses many of the previous concerns. In my view the conclusions drawn are supported enough by the data to warrant publication. The hypothesis concerning the role of TM5 is unique and supported in part through the MD simulations. I would not go as far as saying the data conclusively demonstrate the twisting motion is real, but it is certainly an interesting idea worth proposing. The crystal structure itself is of high quality and shows a unique state for an important model system in drug-proton antiporters. The additional functional data in Figure 4 strengthens the paper and makes the interpretation of the variant data more robust. For sure there are weak spots, as evidenced by the additional MD data that does not behave as their model would predict..

In conclusion this work is of good quality, and worth publishing at this stage.

Reviewer #3 (Remarks to the Author):

The authors have addressed my concerns reasonably.